# Learning to Jump from Pixels

**Gabriel B. Margolis**[1]    **Tao Chen**[1]    **Kartik Paigwar**[2]    **Xiang Fu**[1]

**Donghyun Kim**[1,3]    **Sangbae Kim**[1]    **Pulkit Agrawal**[1]

[1]Massachusetts Institute of Technology    [2]Arizona State University
[3]University of Massachusetts Amherst

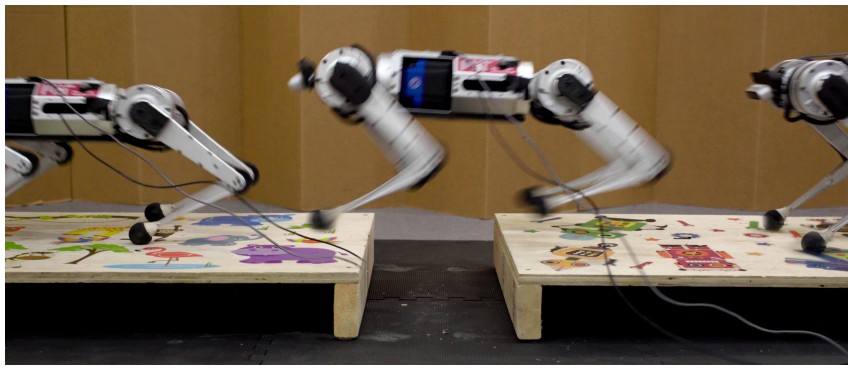

Figure 1: We propose a system architecture called *Depth-based Impulse Control* (DIC) to enable the Mini Cheetah to jump over wide gaps using depth data captured from an onboard camera.

**Abstract:** Today's robotic quadruped systems can robustly walk over a diverse range of rough but *continuous* terrains, where the terrain elevation varies gradually. Locomotion on *discontinuous* terrains, such as those with gaps or obstacles, presents a complementary set of challenges. In discontinuous settings, it becomes necessary to plan ahead using visual inputs and to execute agile behaviors beyond robust walking, such as jumps. Such dynamic motion results in significant motion of onboard sensors, which introduces a new set of challenges for real-time visual processing. The requirement for agility and terrain awareness in this setting reinforces the need for robust control. We present Depth-based Impulse Control (DIC), a method for synthesizing highly agile visually-guided locomotion behaviors. DIC affords the flexibility of model-free learning but regularizes behavior through explicit model-based optimization of ground reaction forces. We evaluate the proposed method both in simulation and in the real world[1].

**Keywords:** Locomotion, Vision, Hierarchical Control

## 1   Introduction

One of the grand challenges in robotics is to construct legged systems that can successfully navigate novel and complex landscapes. Recent work has made impressive strides toward the blind traversal of a wide diversity of natural and man-made terrains [1, 2]. Blind walkers primarily rely on proprioception and robust control schemes to achieve sturdy locomotion in challenging conditions including snow, thick vegetation, and slippery mud. The downside of blindness is the inability to execute motions that anticipate the land surface in front of the robot. This is especially prohibitive on terrains with significant elevation discontinuities. For instance, crossing a wide gap requires the robot to jump, which cannot be initiated without knowing where and how wide the gap is. Without vision, even the most robust system would either step in the gap and fall or otherwise treat the gap as an obstacle and stop. This inability to plan results in conservative behavior that is unable to achieve the energy efficiency or the speed afforded by advanced hardware.

---

[1] Video, code, and appendix available at https://sites.google.com/view/jumpingfrompixels.

5th Conference on Robot Learning (CoRL 2021), London, UK.

State-of-the-art vision-based legged locomotion systems [3, 4, 5, 6, 7, 8, 9, 10] can traverse discontinuous terrain by walking across gaps and climbing over stairs. However, often simplifying assumptions are made in the control scheme such as fixed body trajectory [3], statically stable gait [6, 8], or restricted contact pattern [9, 11]. These assumptions result in conservative and non-agile locomotion. For instance, such systems can walk across small gaps, but cannot jump across big ones.

Planning agile behaviors, such as jumps, on *discontinuous* terrain offers a different and complementary challenge to traversing *continuously uneven* terrain. Executing a jump requires planning the location of the jump, the force required to lift the body, and dealing with severe under-actuation during the flight phase. Past work has demonstrated standing jumps in simulation [12, 13], on a real robot [3], and running jumps in simulation [4, 5, 14, 15, 16]. The most relevant to our work is the demonstration of MIT Cheetah 2 running and jumping over a single obstacle [17]. However, this system was heavily hand-engineered: it assumes straight-line motion, uses a specialized control scheme developed for four manually segmented phases of the jump, and employs a specialized vision system for detecting specific obstacles. Further, the robot was constrained to a fixed gait. Consequently, this system is specific to jumping over one obstacle type, and substantial engineering effort would be required to extend agile locomotion to diverse terrains in the wild.

Traversing discontinuous terrains in more general settings requires a system architecture that can automatically produce a diverse set of agile behaviors from visual observations. To study this problem, we constructed a gap-world environment containing flat regions and randomly placed variable-width gaps. While these environments are much simpler than "in-the-wild", traversing them successfully requires solving many of the core challenges in vision-guided agile locomotion.

Our proposed method, **Depth-based Impulse Control (DIC)**, employs a hierarchical scheme where a high-level controller processes visual inputs to produce a trajectory of the robot's body and a "blind" low-level controller ensures that the predicted trajectory is tracked. This separation eases the task for both the controllers: the high-level is shielded from intricacies of joint-level actuation and the low-level is not required to reason about visual observations, allowing us to easily leverage advances in blind locomotion. Instead of using low-level controllers that track robot's center of mass, a scheme typically known as whole-body control (WBC) [18, 19, 20], we make use of a whole-body impulse controller (WBIC) [21] that reasons about impulses and is therefore appropriate for dynamic locomotion such as jumps. Model-free deep reinforcement learning is used to train the high-level controller that predicts the commands for WBIC from depth images captured from an on-board camera in real-time. We first train our agents in simulation and then transfer them to the real world using the MIT Mini Cheetah robotic platform [22] (Figure 1).

Our overall contribution is a system architecture that enables the robot to: (a) cross a sequence of wide gaps in real-time using depth observations from a body-mounted camera in the real world; (b) requires no dynamics randomization for sim-to-real transfer; (c) does not assume fixed gait and results in emergence of different gaits as a function of robot velocity and task complexity; (d) achieves the theoretical limit of jump width with fixed gaits and even wider jumps with variable gaits and (e) outperforms prior work [3, 8, 23] by making better use of the full range of agile motion afforded by the hardware.

## 2   Method

Our approach, DIC, is guided by the intuition that a wide range of agile behaviors can be generated by using an *adaptive gait schedule* and *commanding the body velocity* of the quadruped. A high forward velocity results in running, whereas different ratios of vertical and forward velocity can control the height and the span of a jump. The adaptive gait schedule allows the robot to change when its foot contacts the ground and thus further expands the range of feasible contact locations and applied forces. As shown in Figure 2, we solve the problem of mapping depth observations to velocity and gait-schedule commands by training a high-level trajectory generator (Section 2.1) with model-free deep reinforcement learning (Section 2.3).

To ensure that the robot tracks these commands, one possibility is to simultaneously train a low-level controller using RL that converts the high-level velocity and gait commands into joint torques. Such a scheme has two drawbacks: (i) sim-to-real transfer issues and (ii) large data requirement for training. Another possibility is to leverage an analytical model of the robot and solve for joint torques using trajectory optimization – a scheme commonly known as whole-body control (WBC) [18, 19,

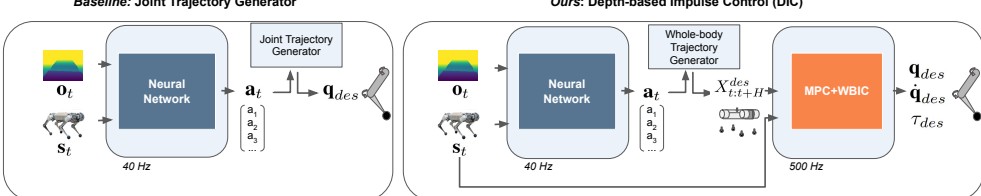

Figure 2: Depth-based Impulse Control (DIC; right) maps robot state and vision to a whole-body trajectory in contrast to previous work that directly predicts joint positions (left). The low-level MPC+WBIC controller enables tracking of highly dynamic whole-body trajectories.

20, 21]. One issue, however, is that a typical WBC tracks the robot's center-of-mass (CoM) [18, 19], which is infeasible during the flight phase of agile motion due to under-actuation of the robot's body. To overcome this issue, we leverage a prior control scheme built on the intuition that changes in body velocity can be realized by modifying the forces applied by the robot's feet on the ground. This frees the controller from the requirement of faithfully tracking the CoM and instead tracks the contact timing and the ground forces applied by the feet. This approach, called whole-body impulse control (WBIC) [21], enables tracking of highly dynamic trajectories set by the high-level controller (see Section 2.2). Our proposed method, Depth-based Impulse Control, integrates WBIC with a vision-aware neural network (Figure 2).

**Whole-body State** The robot's whole-body state at time $t$ is fully defined as

$$X_t = [\mathbf{p}_b, \dot{\mathbf{p}}_b, \ddot{\mathbf{p}}_b, \mathbf{p}_f, \dot{\mathbf{p}}_f, \ddot{\mathbf{p}}_f, \mathbf{C}]_t \in \mathbb{R}^{54} \times [0, 1]^4$$

where $\mathbf{p}_b = [x, y, z, \alpha, \beta, \gamma] \in \mathbb{R}^6$ is the robot body pose (position $(x, y, z)$ and euler angles $(\alpha, \beta, \gamma)$). The terms $\mathbf{p}_f = [p_x^{LF}, p_y^{LF}, p_z^{LF}, p_x^{RF}, p_y^{RF}, p_z^{RF}, p_x^{LR}, p_y^{LR} p_z^{LR}, p_x^{RR}, p_y^{RR}, p_z^{RR}] \in \mathbb{R}^{12}$ denote the position of the Right (R_), Left (L_) Front (_F) and Rear (_R) feet respectively. $\mathbf{C} = [\mathbb{1}_C^{LF}, \mathbb{1}_C^{RF}, \mathbb{1}_C^{LR}, \mathbb{1}_C^{RR}] \in [0, 1]^4$ is the binary contact state of each foot, with $\mathbb{1}_C^f$ taking a value of 1 if foot $f$ is in contact with the ground and a value of zero otherwise.

**Rollout Procedure** The iterative execution routine for our high-level policy and an analytical model-based low-level controller is given by Algorithm 1. The high-level policy $\pi_\theta$ (Section 2.1) selects action $\mathbf{a}_t$, which the whole-body trajectory generator (WTG; Section 2.2) converts to target whole-body trajectory $X_{t:t+H}^{des}$. The low-level controller tracks the whole-body trajectory over horizon $H$ by regulating contact forces. In our experiments, $H = 10$ and the MPC and high-level policy timesteps are 0.036s.

---

**Algorithm 1** Depth-based Impulse Control (DIC)

---

1: $t \leftarrow 0$; $\mathbf{a}_{-1} \leftarrow \mathbf{0}$
2: observe $\mathbf{s}_0, \mathbf{o}_0$
3: **while** not IS-TERMINAL($\mathbf{s}_t$) **do**
4:     sample $\mathbf{a}_t \sim \pi_\theta(\mathbf{a}_t | \mathbf{s}_t, \mathbf{o}_t, \mathbf{a}_{t-1})$
5:     $X_{t+H}^{des} = \text{WTG}(\mathbf{a}_t)$
6:     TRACK-TRAJECTORY($\mathbf{s}_t, X_{t:t+H}^{des}$)
7:     $t = t + 1$
8:     observe $\mathbf{s}_t, \mathbf{o}_t$
9: **end while**

---

## 2.1 High-Level Policy

Let the high-level policy be $\mathbf{a}_t = \pi_\theta(\mathbf{s}_t, \mathbf{o}_t, \mathbf{a}_{t-1})$ where $\mathbf{a}_t$ is the action and $\mathbf{s}_t, \mathbf{o}_t$ denote the robot's internal state and the terrain observation respectively. The action at previous time-step is fed as input to encourage the predicted actions to change smoothly. $\pi$ is represented using a neural network.

**Observation Space** The proprioceptive state $\mathbf{s}_t \in \mathbb{R}^{34}$ consists of the robot body height ($\mathbb{R}$), orientation ($\mathbb{R}^3$), linear velocity ($\mathbb{R}^3$), and angular velocity ($\mathbb{R}^3$), as well as the joint positions ($\mathbb{R}^{12}$) and velocities ($\mathbb{R}^{12}$). The terrain observation $\mathbf{o}_t$ is either a body-centered elevation map $\mathbf{o}_t = \mathbf{E}_t \in \mathbb{R}^{48 \times 15}$ or a depth image $\mathbf{o}_t = \mathbf{I}_t \in \mathbb{R}^{160 \times 120}$ from a body-mounted camera. Observations are normalized using the running mean and the standard deviation.

**Action Space** We train policies with either *fixed*, *variable*, or *unconstrained* gait patterns. In all cases, four continuous-valued dimensions of $\mathbf{a}_t$ encode the target body linear velocity ($\mathbb{R}^3$) and yaw velocity ($\mathbb{R}$). By setting the velocity, we are essentially modulating the target acceleration. For computational efficiency, our low-level controller assumes that the target pitch and roll are near zero, and consequently, we exclude them from the high-level policy output [21]. This assumption does not prevent our system from making agile jumps.

With **fixed gait**, the robot's desired contact state is a cyclic function of time and does not depend on the high-level controller. For instance, the contact schedules for *trot* and *pronk* gaits correspond to:

$$\mathbf{C}_{trot} = \begin{cases} [1,0,0,1] & t < d/2 \mod d \\ [0,1,1,0] & t \geq d/2 \mod d \end{cases} \qquad \mathbf{C}_{pronk} = \begin{cases} [1,1,1,1] & t < d/2 \mod d \\ [0,0,0,0] & t \geq d/2 \mod d \end{cases}$$

where $d$ is the gait cycle duration. In our experiments with fixed gaits, we set $d = 10$. In this scenario, $\pi$ only sets the robot's velocity.

For **variable gait**, the high-level action space is expanded to predict one of the two possible contact states of the feet ($\mathbf{a}_t^c \in [0,1]$). In our setup, *variable pronk* corresponds to choosing one of these states at every time step:

$$\mathbf{C}_{varpronk} = \begin{cases} [1,1,1,1] & \mathbf{a}_t^c = 1 \\ [0,0,0,0] & \mathbf{a}_t^c = 0 \end{cases}$$

We can further relax the assumption about the gait and let the policy choose the contact state for each foot independently ($\mathbf{a}_t^c \in [0,1]^4$) at every time step. We call this **unconstrained gait**, where:

$$\mathbf{C}_{unconstrained} = \big\{[\mathbf{a}_t^c]\big\}$$

determines the contact state of each foot. Flexibility in the contact state allows for *emergence of terrain dependent agile gaits*.

## 2.2   Low-Level Controller

The **Whole-body Trajectory Generator (WTG)** converts action $\mathbf{a}_t$ into an extension of the *desired* whole-body trajectory at time $t + H$, denoted as

$$X_{t+H}^{des} = \text{WTG}(\mathbf{a}_t, X_{t+H-1}^{des}) = [\mathbf{p}_{\mathrm{b}}(\mathbf{a}_t), \dot{\mathbf{p}}_{\mathrm{b}}(\mathbf{a}_t), \ddot{\mathbf{p}}_{\mathrm{b}}(\mathbf{a}_t), \mathbf{p}_{\mathrm{f}}^{\text{raibert}}, \dot{\mathbf{p}}_{\mathrm{f}}^{\text{raibert}}, \ddot{\mathbf{p}}_{\mathrm{f}}^{\text{raibert}}, \mathbf{C}(\mathbf{a}_t)]$$

where the action is converted to a velocity command as $\dot{\mathbf{p}}_{\mathrm{b}}(\mathbf{a}_t) = [\mathbf{a}_t^{\dot{x}}, \mathbf{a}_t^{\dot{y}}, \mathbf{a}_t^{\dot{z}}, \dot{\alpha} = 0, \dot{\beta} = 0, \mathbf{a}_t^{\dot{\gamma}}]$, from which $\mathbf{p}_{\mathrm{b}}(\mathbf{a}_t)$ and $\ddot{\mathbf{p}}_{\mathrm{b}}(\mathbf{a}_t)$ are fixed for consistency with the previous target $X_{t+H-1}^{des}$ assuming linear interpolation between timesteps. The generator computes foot position targets $\mathbf{p}_{\mathrm{f}}^{\text{raibert}}, \dot{\mathbf{p}}_{\mathrm{f}}^{\text{raibert}}, \ddot{\mathbf{p}}_{\mathrm{f}}^{\text{raibert}}$ such that the contact locations satisfy the Raibert Heuristic (Section C.1) and swing trajectories are represented as three-point Bezier curves.

**Whole-body Trajectory Tracking** operates at high frequency with no direct access to terrain information. It consists of a hierarchy of three controllers described in [21] and summarized below:

- A *Model Predictive Controller (MPC)* solves a convex program $\mathbf{f}^{des} = \text{MPC}(X_{t:t+H}^{des}, X_t)$ to convert the desired whole-body trajectory $X_{t:t+H}^{des}$ and current whole-body state $X_t$ into target ground reaction forces $\mathbf{f}^{des}$ for each foot at each timestep. MPC operates at **40 Hz**.
- A *Whole-Body Impulse Controller (WBIC)* applies differential inverse kinematics $\mathbf{q}_{des}, \dot{\mathbf{q}}_{des}, \tau_{des} = \text{WBIC}(X_t^{des}, X_t, \mathbf{f}^{des})$ to find the target position $\mathbf{q}_{des}$, velocity $\dot{\mathbf{q}}_{des}$, and feedforward torque commands $\tau_{des}$ for all joints to optimally track the current step of the the whole-body trajectory $X_t^{des}$ and desired ground reaction forces $\mathbf{f}^{des}$. WBIC operates at **500 Hz**.
- A *Proportional-Derivative Plus Feedforward Torque Controller* takes as input a target position $\mathbf{q}_{des}$, target velocity $\dot{\mathbf{q}}_{des}$, and feedforward torque command $\tau_{des}$ as well as the current position and velocity for each joint. It computes an output torque for each motor at **40 kHz**.

## 2.3   Neural Network Training

**Network Architecture**   The high-level policy $\pi_\theta(\mathbf{a}_t | \mathbf{s}_t, \mathbf{o}_t, \mathbf{a}_{t-1})$ is modeled using a deep recurrent neural network that includes a convolutional neural network (CNN) for processing the raw terrain observation $\mathbf{o}_t$. The output features of CNN are concatenated with proprioceptive inputs $\mathbf{s}_t$, previous action $\mathbf{a}_{t-1}$, and a cyclic timing parameter [23] and passed through a sequence of fully connected layers to output a probability distribution over $\mathbf{a}_t$. Figure 3 illustrates the architecture of the policy network.

**Initialization, Termination, and Reward**
The robot is initialized in a standing position on a terrain with gaps of randomized width and length. An episode terminates if the body or foot positions exceed safety thresholds. The reward $r_t$ at time $t$ is a linear combination of reward for forward progress and penalties on unsafe body and joint states. Further details may be found in appendix B.

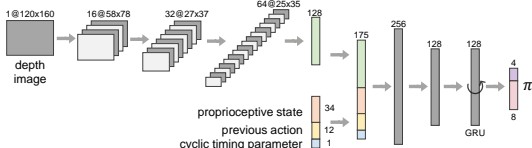

Figure 3: High-level gait prediction network

**Policy Optimization** The parameters of the neural network ($\theta$) are optimized using the PPO [24] algorithm, Adam optimizer [25] with learning rate 0.0003 and batch size 256. During training, 32 environments are simulated in parallel. We find that policies converge within 6000 training episodes, equivalent to 60 hours of simulated locomotion or 12 hours of computation.

**Asymmetric-Information Behavioral Cloning** Learning directly from depth images presents two challenges: (1) *Partial observations*: a front-facing depth camera can only provide information about the terrain in front of the robot, not the terrain underneath its feet, making the contact-relevant terrain partially observed. (2) *Sensory variance*: the depth image obtained from a body-mounted camera is dependent on the robot pose. This introduces variance in perception across trials, even when the robot is traversing the same terrain.

Variance makes learning more challenging, and partial observations necessitate the use of a recurrent network architecture. These factors make learning directly from depth images less sample-efficient than learning from heightmaps. In addition, rendering depth images is more computationally expensive than cropping heightmaps, which makes learning from depth images less wall-clock efficient.

To address these challenges, we propose a two-stage approach that first trains an expert policy ($\pi_\text{E}$) with privileged access to a ground-truth heightmap. A second student policy ($\pi_\text{BC}$) is trained from depth inputs to mimic the expert policy. For this, we use a variant of Behavioral Cloning (BC) known as DAgger [26] to minimize the KL-divergence between the output action distribution of the imitating agent $\pi_\text{BC}(a|s)$ and the expert $\pi_\text{E}(a|s)$: $\min D_\text{KL}\big(\pi_\text{E}(a|s)||\pi_\text{BC}(a|s)\big)$. Prior works have applied behavioral cloning from privileged information to other settings [1, 27].

## 3 Experimental Setup

**Hardware**: We use the MIT Mini Cheetah [22], a 9kg electrically-actuated quadruped that stands 28cm tall with a body length of 38cm. A front-mounted Intel RealSense D435 camera provides real-time stereo depth data and an onboard computer [3] run the trajectory-tracking controller described in Section 2.2. Data from the depth camera is processed by an offboard computer that communicates the output of the high-level policy to the robot via an Ethernet cable.

**Simulator**: We train high-level policy using the PyBullet [28] simulator. To obtain data from the mounted depth camera, we use a CAD model of our robot and sensor's known intrinsic parameters.

**Gap World Environment**: To evaluate the ability of our system to dynamically traverse discontinuous terrains, we define a test environment consisting of variable-width gaps and flat regions. The difficulty of traversing gap worlds depends on the proximity of gaps as well as gap width, with closer and wider gaps presenting a greater challenge to the controller. Our training dataset consists of randomly generated gaps with uniform random width between $W_\text{min} = 4$ and $W_\text{max} \in [10, 20, 30]$ centimeters, separated by flat segments of randomized width 0.5 to 2.0 meters. Our test dataset contains novel terrains drawn from the same distribution.

**Baselines**: We compare our method to a model-free baseline, Policies Modulating Trajectory Generators, and a model-based baseline, Local Foothold Adaptation. For details of these baselines, refer to Appendix D, E.

## 4 Results

### 4.1 Simulation Performance

**Fixed Gait** We train Depth-based Impulse Control to cross gaps using trotting and pronking gaits. For both trotting and pronking, our visually-guided approach succeeds at above 90% of gap cross-

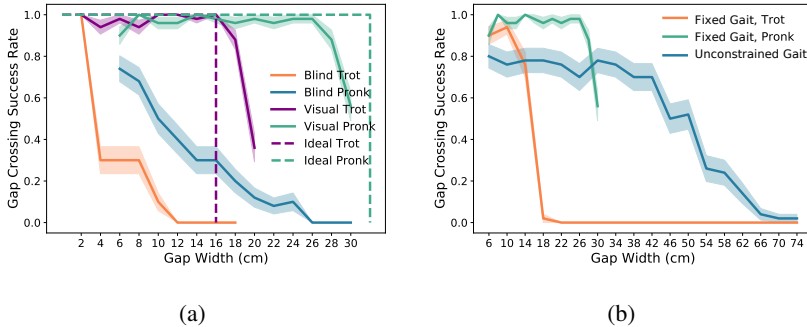

(a)                                                  (b)

Figure 4: **(a)** Visually guided fixed gait policies significantly outperform blind policies and are close to the "ideal" theoretical limit. Shaded regions indicate standard error of the mean. **(b)** A comparison of performance among policies trained with fixed gait and unconstrained gait demonstrates the flexibility and dynamic range of our method.

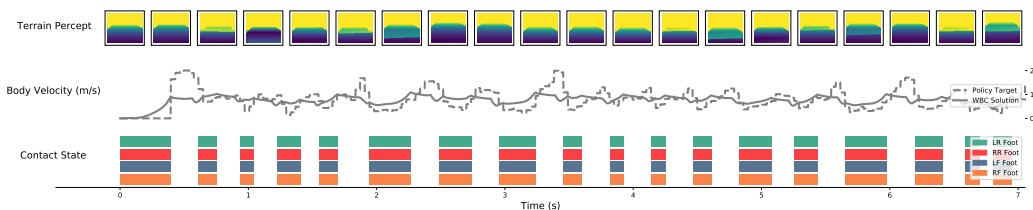

Figure 5: Contact schedule generated by our variable gait policy. Given a terrain observation (top), the policy modulates body velocity (middle) and contact duration (bottom) to traverse 30cm gaps.

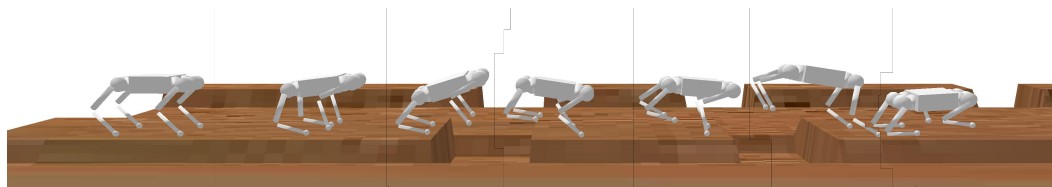

Figure 6: In simulation, our unconstrained gait policy can traverse gaps up to 66cm wide.

ing attempts up to the theoretical limits derived in the supplementary material (Section C). Figure 4a reports the performance of our method relative to this theoretical limit. Ideal performance is derived from maximum stride length given velocity, foot placement, and contact schedule constraints. Note that while the theoretical limits are derived assuming zero yaw, the learned trotting controller learns to move with nonzero yaw, thus extending the foot placements further apart and beating the ideal. Our method also outperforms blind locomotion (Figure 4a) and a Local Foothold Adaptation baseline [3] (Figure D.2), particularly on large gaps.

**Unconstrained Gait**  We relax all constraints on contact schedule and train a controller with a *vision-adaptive contact schedule* to cross wide gaps. Figure 4b reports the performance of unconstrained gait gap crossing in simulation. Unconstrained gait policies outperform those with fixed gait, crossing gaps that are much wider. When trained with extremely wide (40- to 70-cm gaps), DIC learns to select a variable-bounding contact schedule which achieves superior performance to trotting and pronking for very large gaps (Figure 6). When we restrict the maximum gap size to 40cm or less, a variable-timing pronking gait emerges in the unconstrained gait controller. Figure 5 illustrates the variable contact timings and velocity modulation of the variable pronking controller in simulation. Similar to concurrent work [29] which has demonstrated the emergence of variable gaits for energy minimization on flat ground; we observe emergent gait adaptation for safe traversal of discontinuous terrain.

**Ease of Training**  Our method successfully navigates gaps of different width with different gaits using the same reward function and trajectory generator structure. In contrast, we found that the PMTG baseline was highly sensitive to the tuning of the reward and trajectory generator for each gait and environment. We first tuned the trajectory generator, residual magnitudes, and reward function

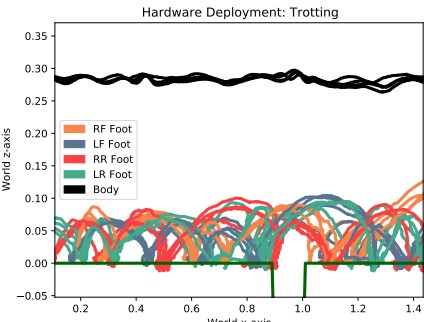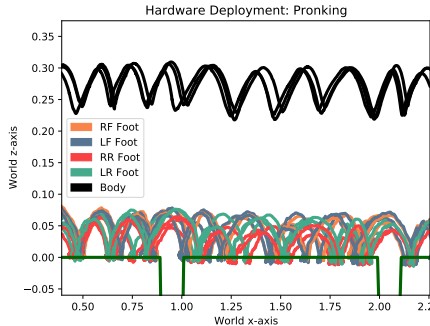

Figure 7: Motion capture data verifies the transfer of planned trajectories to the hardware system. The contact sequence varies across trials, but adapts to avoid the illustrated gaps each time.

of PMTG for sim-to-real forward locomotion on flat ground; details and video of the baseline can be found at the project website[1]. We found the parameters that succeeded at sim-to-real on flat ground were prohibitively conservative and failed to learn any gap-crossing behavior when the maximum gap width $W_{max}$ was 10cm for trotting or 20cm for pronking. To overcome this issue, we applied specialized reward design and expanded the range of the trajectory generator parameters. While the re-tuned agent was able to cross gaps longer than the aforementioned range, the resulting behaviors overrode the TG with irregular gaits indicative of simulator exploitation.

## 4.2 Real World Performance

**Deployment** We deploy DIC in fully real-time fashion on the MIT Mini Cheetah robot [22], directly making use of depth images and an onboard state estimator. In this setting, we record successful gap crossings up to 16cm. We refer the reader to the project website for video evaluation[1].

To study the impact of sensor noise on transfer, we also deploy DIC using ground-truth state information via motion capture and terrain heightmap. With these adjustments, we are able to consistently cross gaps up to 26cm on the real robot. Figure 7 plots motion capture data from three such deployments each for adaptive trotting (left) and adaptive pronking (right). The relevant cross-section of the terrain surface is drawn in dark green. Although the foot placements of the robot differ across runs due to noise in the system dynamics, DIC adapts to avoid stepping in a gap in each case.

From these experiments, we identify two main challenges which prevent our method from transferring for wider gaps: (i) drift in state estimation caused by sensor noise and imprecise knowledge of contact timing; (ii) violation of the assumption made by the low-level controller that the robot's feet do not slip while in contact with the floor, especially during aggressive motion. We refer the reader to the project website for video of example failure cases[1].

## 4.3 Vision and Behavioral Cloning

**Behavioral Cloning (BC)** Table 1 illustrates that behavioral cloning from heightmaps to depth images offers an advantage over learning directly from depth images in most cases after 10M training steps and 1M cloning steps. We note that cropping heightmaps is faster than rendering depth images, resulting in an additional wall-clock time benefit to BC. These results also demonstrate that the combination of behavioral cloning with a variable gait schedule is beneficial, with the cloned Variable Pronk achieving highest performance for wide gaps of any fixed or variable gait policy.

**Recurrent Architecture** We find that student policies with recurrent architecture consistently yield higher final performance than without, particularly for environments with larger gaps which require more dynamic motion (Table 1). This suggests that the hidden state is helpful in forming a useful representation of unobserved terrain regions given the observation history.

## 5 Related Work

**Model-free RL for locomotion** is shown to benefit from acting over low-level control loops rather than raw commands [30]. Robust walking methods including RMA [2, 31] as well as recent work

Table 1: Gap crossing success rate for RL policies (with Trotting (T), Pronking (P), or Variable Pronking (VP)) trained on various maximum gap widths with with height maps, depth images as input respectively, and the policy produced by behavioral cloning with and without recurrent architecture. For model trained with maximum gap width $W_{\max}$, the evaluated gap width is $W_{\max} - 5$.

| Input | T, 10cm | T, 20cm | P, 20cm | P, 30cm | VP, 30cm |
|---|---|---|---|---|---|
| *Heightmap (MLP)* | 1.0 | 1.0 | 1.0 | 0.7 | 1.0 |
| *Depth Image (RNN)* | 0.6 | 0.3 | 0.9 | **0.9** | 0.7 |
| *Heightmap (MLP)* → *Depth Image (MLP)* | 1.0 | 0.9 | 0.1 | 0.0 | 0.0 |
| *Heightmap (MLP)* → *Depth Image (RNN)* | **1.0** | **1.0** | **1.0** | 0.4 | **1.0** |

on ANYmal [1, 32] and Cassie [33] learn conservative, vision-free policies to predict joint position targets for a PD controller and achieve sim-to-real transfer using a combination of reward shaping, system identification, domain randomization, and asymmetric-information behavioral cloning. Previous work in simulation [5, 13, 15] has applied model-free reinforcement learning to traversal of discontinuous terrains in simulation. [5] notably applied model-free RL to the problem of crossing stepping stones with physically simulated characters, but this method did not use realistic perception or take measures to promote sim-to-real transfer.

**Model-based control for locomotion** has achieved highly dynamic blind walking [34], running [21], and jumping over obstacles [17] using known quadruped whole-body and centroidal dynamics. Other works have applied model-based control to terrain-aware navigation of a mapped environment, typically with complete information about the terrain [8, 35]. In general, control strategies based on known models are high-performing and robust where the state is known and the model is sufficiently accurate. In contrast, model-free controllers excel at incorporating unstructured or partially observed state information when large data is available.

**Interfacing Model-based and Model-Free Methods**. A previous line of work has leveraged model-free perception for foothold selection. [11] locally adapted foot placements to safe footholds predicted by a CNN. RLOC [10] similarly uses a learning-based online footstep planner in combination with a learning-modulated whole-body controller to perform terrain-aware locomotion. Unlike our method, [10] uses a complete terrain heightmap as observation, plans by targeting foot placements, and is limited to relatively conservative fixed walking and slow trotting gaits. On the other hand, concurrent work applies RL to modulate a model-based controller's target command without perception. [36, 29] demonstrated that using a model-free policy to choose contact schedules for a reduced-order model leads to the emergence of efficient gait transitions during blind flat-ground locomotion. [37] demonstrates the integration of a model-free high-level controller with a centroidal dynamics model. This framework deployed with a fixed trotting gait is demonstrated to achieve flat-ground and conservative terrain-aware locomotion. Unlike our work, [37] does not demonstrate gaits with flight phases or plan from realistic terrain observations.

## 6 Conclusion and Discussion

We have presented a vision-based hierarchical control framework capable of traversing discontinuous terrain with gaps. The combination of model-free high-level trajectory prediction and model-based low-level trajectory tracking enables us to simultaneously achieve high performance and robustness.

While our system advances the state-of-the-art, there are many avenues for improvement. First, while we are able to train policies that can jump gaps as long as 66 centimeters in simulation, we can only transfer to gaps up to 26 centimeters in the real world. We identify a few obstacles to transfer in Section 4.2 and further note the limitation that the contact force optimizer does not account for robot's kinematic configuration, sometimes resulting in infeasible or overly conservative target impulses. Finally, while we only present results in the gap-world environment, our approach may be simply extended to combinations of rough continuous terrains and additional classes of discontinuous terrains such as stairs. We leave such experimentation to future work.

**Acknowledgments**

The authors acknowledge support from the DARPA Machine Common Sense Program. This work is supported in part by the MIT Biomimetic Robotics Laboratory and NAVER LABS. We are grateful to Elijah Stanger-Jones for his support in working with the robot hardware and electronics. The authors also acknowledge MIT SuperCloud and the Lincoln Laboratory Supercomputing Center for providing HPC services.

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
