# OpenReview forum: "Learning to Jump from Pixels"
_robot-learning.org/CoRL/2021/Conference — CoRL2021 Poster_

### Official Review · Reviewer_dwZr · 2021-07-23

**Originality:** Very Good
**Technical Quality:** Good
**Clarity Of Presentation:** Good
**Impact:** 4

**Recommendation:**

Weak Accept: I recommend accepting the paper, but will not argue for my recommendation if the majority of other reviewers have a different opinion.

**Summary:**

In this work, the authors proposed a novel architecture to train locomotion policies that can solve challenging visual locomotion tasks such as gap crossing. The controller contains a high level, learned policy that can modulate the desired body pose and velocity of the robot based on visual inputs (height map or depth map), and a low level MPC-WBC based motion controller to track the desired center of mass pose overtime. The training contains two stages. In the first stage, a teacher policy is trained using reinforcement learning and using ground truth height maps. In the second stage, a student policy, which takes raw depth image as inputs, is trained through behavior cloning. Using the learned policy, Mini Cheetah can trot or pronk over wide gaps in simulation. The authors also deployed the teacher policy to the real robot using ground truth height maps, and demonstrated great sim-to-real transfer results.

**Issues:**

Please see my comments on the main limitations of the paper.

On top of that, I think Table 1 can use more baseline results. Also, I think it is worthwhile to study the asymptotic performance of RNN only policy (RL trained). The RNN policy is expected to be much harder to train given the limited information, so naturally it can take more samples to reach the same performance. Since there is no training curve provided I am not even sure if the RNN policy is converged. Need more details on this.

Another question I have is the quality of depth image. From the video the depth images look surprisingly good, which contradicts from my experiences with D435 cameras: There is no invalid band, no noise near the edges, etc. More discussions are needed on this.

**Reviewer Expertise:**

Excellent: Expert knowledge on the topic of the paper

**Strengths And Weaknesses:**

The main strengths of the paper are:

(1) A novel architecture to combine learning with traditional control methods. Using a learned "planning" policy to generate reference center of mass motions and track the motion using a traditional, model predictive control based controller. This will minimize the sim to real gap especially for dynamic locomotion gaits such as pronking.

(2) A two-stage training with teacher and student policies. The teacher can solve the easier task with more info, i.e. the ground truth height map, and the student can mimic the teacher's behavior with limited inputs (front view depth images) and a RNN based memory module. This will increase the learning efficiency.

(3) Good sim-to-real transfer results of the teacher policy. The authors demonstrated dynamical pronking over a 30 cm wide gap which is a pretty significant result.


The main limitations of the paper are:

(1) Not enough baselines to compare with. In many non-learning based papers,  there is this heuristic baseline where the foothold choice is based on the projected landing position of the feet:  A local optimization is used to search valid, nearest foothold location if the projected landing position is inside the gap. Consider adding this as a baseline to compare with at least the teacher policy proposed in this paper.

(2) Not enough real robot results for the student policy (i.e. depth RNN). The authors demonstrated successful sim-to-real transfer to the teacher policy which uses ground truth height maps. However, the results from the depth RNN policy is not reported. Even if there are difficulties in sim-to-real transfer of the student policy, it is still worth reporting and discussing the results. If the transfer fails, is it because of the state estimation, or because of the imperfect ego-motion model embedded in the RNN, or discrepancies in the visual inputs (i.e. distribution shift).

**Summary Of Recommendation:**

How to elegantly use visual info in locomotion to solve challenging tasks is a hot and challenging topic. Based on my assessments of the strengths and limitations of the paper, I recommend accepting the paper conditioned on addressing my comments.

---

> ### Author Response · Authors · 2021-08-25
> **Response to Reviewer dwZr**
>
> We thank reviewer dwZr for their constructive feedback. We will address the reviewer's concerns as follows:
>
> **"In many non-learning based papers, there is this heuristic baseline where the foothold choice is based on the projected landing position of the feet: A local optimization is used to search valid, nearest foothold location if the projected landing position is inside the gap. Consider adding this as a baseline to compare with at least the teacher policy proposed in this paper."**
>
> **RESPONSE:** Deriving an upper bound on the performance of the local foothold adaptation baseline (Appendix B), we find that our method significantly outperforms the best possible performance of an ideal foothold adaptation approach. Based on the implementation of search-based foothold adaptation from [1], we have quantified this performance difference in Appendix Figure B.2. It is also worth noting that foothold adaptation is complementary to our approach, so it is possible in the future to combine the two techniques to further improve performance.
>
> **"Not enough real robot results for the student policy (i.e. depth RNN)."**
>
> **RESPONSE:** We are happy to report that we have now successfully deployed the student policy with real camera input in the loop. The manuscript has been updated to reflect this, and the relevant video and analysis of failure modes have been added to the website here: [https://sites.google.com/view/jumpingfrompixels/transfer-experiments](https://sites.google.com/view/jumpingfrompixels/transfer-experiments).
>
> **"From the video the depth images look surprisingly good, which contradicts from my experiences with D435 cameras: There is no invalid band, no noise near the edges, etc. More discussions are needed on this."**
>
> **RESPONSE:** To avoid the invalid band of our stereo depth sensor, we crop the depth image on the left by a fixed amount before passing it to the controller. We then apply standard hole-filling and threshold filters to reduce noise in the image, yielding the result shown in the video. We have added a complete description of our heightmap preprocessing as Appendix Section E in the supplementary material. Figure E.7 illustrates example depth images before and after preprocessing.
>
>
> **REFERENCES:**
>
> 1. D. Kim, D. Carballo, J. Di Carlo, B. Katz, G. Bledt, B. Lim, and S. Kim. Vision aided dynamic exploration of unstructured terrain with a small-scale quadruped robot. In IEEE International Conference on Robotics and Automation, 2020.

---

### Official Review · Reviewer_NUCw · 2021-07-23

**Originality:** Fair
**Technical Quality:** Fair
**Clarity Of Presentation:** Poor
**Impact:** 2

**Recommendation:**

Weak Reject: I recommend rejecting the paper, but will not argue for my recommendation if the majority of other reviewers have a different opinion.

**Summary:**

This paper considers the problem of controlling a legged robot to jump over gaps. In its particular problem setting, a robot has no access to an elevation map. The paper proposes a hybrid controller to perform jumping maneuvers with an RNN of image input. The proposed algorithm is evaluated on a physical walking robot with the rate of successful jumps given different kinds of input.

**Issues:**

1. Figure 1 violates formatting guidelines, because it is placed before the abstract.
2. Writing needs to be improved. See list of weaknesses.
3. What should be taken away from Figure 5? All the feet are on the ground at the same time? How can that be?

**Reviewer Expertise:**

Good: General knowledge of the area

**Strengths And Weaknesses:**

*Strengths*
* The paper seems to address a challenging problem with significance to the robotics community.
* Experiments use data from a physical robotic system.

*Weaknesses*
* The paper is more of a technical report than a report of scientific research.
* The writing makes it difficult to identify the research questions.
* The writing makes it difficult to find and follow the paper's main narrative.
* The writing makes several vague and overly-general claims---these should be sharpened. For instance, the paper claims RL is data-inefficient (line 52). This statement is too vague to be meaningful. Furthermore, this statement seems generally false because some RL algorithms are quite efficient relative to their alternatives. Overall, the writing needs to reduce these instances of loose claims and make a more specific points.
* Line 61: Remove exclamation point.
* Algorithm 1 is not very informative. All the novel details one may need to implement and understand this work is in TRACK-TRAJECTORY().
* Lines 41--43: This is a subjective claim; it should be removed or edited.
* Behavioral cloning and the Sim-to-real transfer bits seemed strange to include in this study. Can the authors expand on why they think these parts are important for their study?


**Summary Of Recommendation:**

Currently I am leaning to reject this paper because of concerns about its significance and correctness. The paper seems like it was put together in a rush. This made the writing difficult to parse, and it made it challenging to identify what research was being done---what the questions were and how the experiments shed light onto these issues.

I can tell that the paper is trying to tackle a difficult control problem, and I appreciate how its experiments went the extra distance to deploy its proposed algorithm onto a physical robot. Unfortunately, because of issues with clarity and execution, those results appear somewhat inconclusive. How do the performance differences between baselines target the concerns mentioned in the introduction? For example, learning to jump without an elevation map. Is the conclusion that learning to jump can be sufficiently done with an RNN? Besides the results, the experiments do not contain an uncertainty analysis that would give readers confidence in their reproducibility.

I am open to adjusting my score if perhaps, during the rebuttal, the authors can clarify their position and expand on their results.

---

> ### Author Response · Authors · 2021-08-25
> **Response to reviewer NUCw (1/2)**
>
> We thank reviewer NUCw for their thoughtful feedback. We will address the reviewer's concerns as follows:
>
> **"The writing makes it difficult to identify the research questions... The writing makes it difficult to find and follow the paper's main narrative... perhaps, during the rebuttal, the authors can clarify their position and expand on their results."**
>
> **RESPONSE:** We are sorry that our writing was unclear. We have updated the manuscript based on the reviewer's writing-related suggestions.
>
> Main Narrative: In this work, we consider the task of dynamically traversing highly discontinuous terrains with gaps. Unlike uneven but continuous terrain, which is the subject of much previous work [1, 5], discontinuous terrain requires the incorporation of vision since large regions of the terrain (such as gaps) are unsafe to step onto. Additionally, discontinuous terrain necessitates a different set of dynamic maneuvers than uneven but continuous terrain. The overarching question we address is controller design for quadruped locomotion on discontinuous terrain. The concrete questions we address are: What components of the controller should be analytically modeled, and how should we model them? What components of the controller can be learned, and how can we learn them? How can the learnt and analytical components be put together to maximize performance?
>
> Toward this broad goal, we conduct experiments that evaluate the following specific research questions:
>
> - Q1: How do our modeling and controller design choices impact locomotion performance across discontinuous terrain? A1: We propose a novel architecture combining whole-body impulse control with model-free learning. We demonstrate that this approach achieves theoretical limits for the challenging task of locomotion on gapped terrain and outperforms a fully model-based alternative (Fig 4, Appendix B). We also empirically demonstrate better performance than a completely model-free system (Appendix C, D).
> - Q2: Prior works in visual locomotion use heightmaps, which must either be pre-constructed or mapped online introducing complexity and latency. Can our learning-based approach use egocentric depth images instead of heightmaps to enable simple and real-time deployment? A2: Yes, but the naive application of the same approach used for heightmaps is insufficient. The application of behavioral cloning and the use of a recurrent architecture are necessary for this setup, as demonstrated by ablation in Table 1.
> - Q3: Our innovations in Q1 and Q2 respectively offer new approaches to robust dynamic control and real-time visual adaptation. Can our novel pipeline enable state-of-the-art behavior on a real robot? A3: Positive sim-to-real results (Figure 6, Video Supplement, [https://sites.google.com/view/jumpingfrompixels/transfer-experiments](https://sites.google.com/view/jumpingfrompixels/transfer-experiments)) indicate yes.
>
> **"Behavioral cloning... seemed strange to include in this study. Can the authors expand on why they think these parts are important for their study?"**
>
> **RESPONSE:** Our work takes a step towards legged robots that can adapt to diverse and novel terrains using vision, incorporating vision in real-time without the bottleneck of heightmap reconstruction. Cited prior works in visual locomotion [1, 2, 4] present controllers which reason from heightmaps or parametric terrain information. These intermediate representations must either be pre-constructed or mapped online, introducing complexity and latency to the system.
>
> To overcome these issues, one must train policies directly from egocentric depth images directly available from RGBD sensor on the robot. However, ablation in Table 1 shows that training a policy directly with depth images yields lower performance than training with heightmaps. We demonstrate that learning is easier as a two-stage process, in which a teacher policy is first trained from heightmaps and then a student policy learns to mimic its behavior using depth images.
>
> (continued)

---

> > ### Author Response · Authors · 2021-08-25
> > **Response to reviewer NUCw (2/2)**
> >
> > (continued)
> >
> > "**the Sim-to-real transfer bits seemed strange to include in this study. Can the authors expand on why they think these parts are important for their study?"**
> >
> > **RESPONSE:** Simulation is imperfect. Consequently, there is a large gap in the locomotion literature between the state of the art in simulation and the state of the art on a real robot. Purely learning-based approaches [1, 2] have succeeded at gap-crossing tasks in simulation, but have not demonstrated results on real hardware. Since our method involves training in simulation, we think it is critical to demonstrate results on the real robot to establish our method's credibility. Therefore, sim-to-real transfer is necessary.
> >
> > **"Besides the results, the experiments do not contain an uncertainty analysis that would give readers confidence in their reproducibility."**
> >
> > **RESPONSE:** Thank you for bringing this concern to our attention. Sources of uncertainty in our system's performance might include the selection of random seeds during training and the randomized generation of test terrains during evaluation.
> >
> > To quantify the reproducibility of the learning procedure, we will conduct five random seed trials for each entry in Table 1 and update this table to report the mean and standard deviation among them. Our experience with the system and initial results suggest that the learning procedure has low sensitivity to random seed selection.
> >
> > We have added shaded regions representing standard error to the performance results in Figure 4. With each data point representing the outcome of locomotion across 50 randomly sampled terrains, the standard error in performance measurement is low. Terrains are sampled with randomized gap width and gap spacing as described in section 4 (Gap World Environment).
> >
> > **"What should be taken away from Figure 5? All the feet are on the ground at the same time? How can that be?"**
> >
> > **RESPONSE:** Yes, all four feet are on the ground at the same time in Figure 5. This gait pattern, "pronking", is established in prior locomotion work [3], although it is challenging to achieve under conservative control frameworks or on larger, heavier robots. The robot is shown executing this gait in Figure 1 of our paper. The supplementary video introduces this gait by name at time 1:34. Video of our method executing this gait is in the supplementary video from time 5:04-5:23.
> >
> > **REFERENCES**
> >
> > 1. V. Tsounis, M. Alge, J. Lee, F. Farbod, and M. Hutter. Deepgait: Planning and control of quadrupedal gaits using deep reinforcement learning. In IEEE International Conference on Robotics and Automation, 2020.
> > 2. Z. Xie, H. Y. Ling, N. H. Kim, and M. van de Panne. Allsteps: Curriculum-driven learning of stepping stone skills. Computer Graphics Forum, 39(8):213–224, 2020.
> > 3. D. Kim, J. D. Carlo, B. Katz, G. Bledt, and S. Kim. Highly dynamic quadruped locomotion via whole-body impulse control and model predictive control. ArXiv, abs/1909.06586, 2019.
> > 4. P. Fankhauser, M. Bjelonic, C. D. Bellicoso, T. Miki, and M. Hutter. Robust rough-terrain locomotion with a quadrupedal robot. In 2018 IEEE International Conference on Robotics and Automation (ICRA), pages 1–8. IEEE, 2018.
> > 5. C. D. Bellicoso, C. Gehring, J. Hwangbo, P. Fankhauser, and M. Hutter. Perception-less terrain adaptation through whole body control and hierarchical optimization. In 2016 IEEE-RAS 16th International Conference on Humanoid Robots (Humanoids), pages 558–564. IEEE, 2016.

---

### Official Review · Reviewer_8yhZ · 2021-07-27

**Originality:** Good
**Technical Quality:** Very Good
**Clarity Of Presentation:** Very Good
**Impact:** 3

**Recommendation:**

Weak Accept: I recommend accepting the paper, but will not argue for my recommendation if the majority of other reviewers have a different opinion.

**Summary:**

This paper presents an approach to train control policies to execute agile gap jumping. A combination of reinforcement learning and model-based whole body control is used to generate robust policies. Policies is first trained with height map then distilled into a policy that takes raw camera images as input. Results demonstrate agile jumping motions across gaps, both in simulation and on the real robot.

**Issues:**

1. As mentioned above, it is not clear how foot placement strategy plays a role here, as one challenge of the tasks is precise foot placement. Please provide additional information about how this is implemented. My understanding is that a Raibert heuristic is used, but if the Raibert heuristic generates a target into the gap, how is this being handled? Especially in the case of using pure pixel, where gap avoidance becomes impossible?

2. While ablations show that learning contact scheduling is important, it is not clear why learning target body velocity is also important. Is it possible to just use the some reasonable fixed desired body pose/velocity? e.g, desired body height being the nominal height and desired body x, y velocities being some reasonable values.

3. In the Raibert heuristic (line 427 in appendix), how is the duration of the next placement calculated given that the contact schedule is not fixed?

4.  In the video, all scenarios seem to have gap width less than the body length (38cm). It will be interesting to see the larger gap that the gait-free policy is trained with, even if it is mostly failure.

**Reviewer Expertise:**

Very good: Comprehensive knowledge of the area

**Strengths And Weaknesses:**

Pro:

1. A system for training agile locomotion skills across gap that combines reinforcement learning and model-based control, with demonstration on the real robot.

Con:

1. One important/difficult part of gap crossing is the requirement of precise foot placement to avoid gap. It is not clear in the paper how this is being handled.

2. While ablations show the importance of learning contact scheduling, it is not clear why learning body velocity is necessary.

**Summary Of Recommendation:**

This paper presents an approach to generate control policies for a quadruped to jump across terrains with gaps. The techniques look sounds, given that it work well both in simulation and on the real robot. But some clarifications are needed (see issues section).

However, given that the title is "learning to jump from pixels" and this is a robotics conference, one would imagine a system that can enable jumping on real robot with real camera input, which is not the case in this paper.

That being said, given the preliminary real robot experiment and a system that works well in simulation, this paper still provides a good initial step in training agile robot motion with perception.

---

> ### Author Response · Authors · 2021-08-25
> **Response to reviewer 8yhZ**
>
> We thank reviewer 8yhZ for their useful and encouraging feedback. We will address the reviewer's concerns as follows:
>
> **"One would imagine a system that can enable jumping on real robot with real camera input, which is not the case in this paper."**
>
> **RESPONSE:** We have now deployed the learned policy on the real robot with real camera input and onboard state estimator (not motion capture). We found that locomotion was stable and appropriate adaptive behavior was generated in response to gaps for trotting and pronking gaits. Due to limited lab access, we have not yet deployed the variable-gait policy in this setting. The manuscript has been updated to reflect this, and the relevant video and analyses have been added to the supplementary materials as well as the website here for easy access: [https://sites.google.com/view/jumpingfrompixels/transfer-experiments](https://sites.google.com/view/jumpingfrompixels/transfer-experiments).
>
> **"In the video, all scenarios seem to have gap width less than the body length (38cm). It will be interesting to see the larger gap that the gait-free policy is trained with, even if it is mostly failure."**
>
> **RESPONSE:** We have added a video and trajectory visualization of the simulated robot crossing 50-centimeter gaps to the supplementary materials as well as the website here for easy access: [https://sites.google.com/view/jumpingfrompixels/long-jumping](https://sites.google.com/view/jumpingfrompixels/long-jumping). Since the robot moves with a higher velocity during traversal of these large gaps, additional safety considerations will be necessary for our lab space, and deployment on the real robot is still a work in progress.
>
> **"It is not clear how foot placement strategy plays a role here, as one challenge of the tasks is precise foot placement... While ablations show that learning contact scheduling is important, it is not clear why learning target body velocity is also important."**
>
> **RESPONSE:** The reviewer correctly notes that our stepping sequences strictly follow the Raibert heuristic, which fixes the foot placement locations given the robot body velocity and contact timing (Appendix B). If the Raibert heuristic will result in placing a foot in a gap, there are two possible ways to change this:
>
> *Foothold Adaptation*: Adjust the foot placement while maintaining the same velocity and contact schedule. This results in a deviation from the Raibert heuristic.
>
> *Pose Adaptation:* Do not deviate from the Raibert heuristic, but adjust the robot body velocity and/or contact timing itself. For example, if the forward velocity of the body increases, the distance between foot placements will increase under the Raibert heuristic, even when the contact timings remain the same. By slowing down or speeding up when it sees an approaching gap, our robot can set itself up for a smooth and powerful motion that maintains the Raibert heuristic and does not step in a gap.
>
> Some cited prior works [1, 2] use the foothold adaptation approach to avoid placing footholds on patches of continuous, uneven terrain that are classified as unsafe. However, as the reviewer correctly points out, foothold adaptation is easiest to apply when not working directly from pixels. In addition, as we show in our revised Appendix B, the foothold adaptation baseline cannot perform as well as our approach for crossing large gaps. Our work exclusively focuses on the development of body velocity and contact schedule adaptation.
>
> We have found that adapting the velocity and contact schedule while fully adhering to the Raibert heuristic can give rise to useful, dynamic behaviors with sufficient precision to traverse gaps.  In the future, foothold adaptation could be combined with our approach to velocity and contact schedule adaptation, and we expect that this would produce a more versatile system.
>
> **"In the Raibert heuristic (line 427 in appendix), how is the duration of the next placement calculated given that the contact schedule is not fixed?"**
>
> **RESPONSE:** The policy selects the contact schedule for some horizon H into the future. The result is that the duration of the next placement is almost always already specified within this horizon when the Raibert heuristic is calculated. In the case that the duration is not yet known, we use a fixed default value for heuristic calculation, which is updated when the true value becomes available.
>
> **REFERENCES:**
>
> 1. D. Kim, D. Carballo, J. Di Carlo, B. Katz, G. Bledt, B. Lim, and S. Kim. Vision aided dynamic exploration of unstructured terrain with a small-scale quadruped robot. In IEEE International Conference on Robotics and Automation, 2020.
> 2. O. A. V. Magana, V. Barasuol, M. Camurri, L. Franceschi, M. Focchi, M. Pontil, D. G. Caldwell, and C. Semini. Fast and continuous foothold adaptation for dynamic locomotion through cnns.IEEE Robotics and Automation Letters, 4(2):2140–2147, 2019.

---

> ### Comment · Reviewer_8yhZ · 2021-08-31
> **happy with the response**
>
> I am happy with the response from the authors as well as the additional experiments, so I will keep my positive rating.

---

### Meta-Review · Area_Chair_LpBE · 2021-08-11

**Recommendation:** Accept (Poster)
**Confidence:** 4

**Metareview:**

This paper proposes to combine model-free learning and model-based control to enable a quadruped robot to jump over gaps using visual inputs. All reviewers agree that the paper is tackling an important and challenging control problem that could have potentially large impact. The combination of reinforcement learning and MPC is neat and could inspire follow-up works. The revision has significantly improved the quality of this paper. Original concerns about the clarity of writing, lack of baselines, and missing on-robot experiments using the student policies are sufficiently addressed in the revision. New experiments on the robots are impressive.

The PMTG baseline for gap-crossing in simulation looks a bit weird though: many times the robot steps into the gap but seems to still get the contact force to keep it balanced (e.g. 00:06s in Trotting: Relaxed TG, rear left leg; also in the first plot Trajectory Plots). This does not change any conclusion of the paper because the PMTG baseline would perform even worse. But explaining/fixing this artifact will further improve the paper quality.

---

> ### Author Response · Authors · 2021-08-25
> **Response to Area Chair LpBE**
>
> We thank the AC for the metareview. We have addressed the reviewers' major concerns by making following revisions:
>
> - Added Foot Placement Adaptation baseline and demonstrated that our approach performs better.
> - Added real-world robot experiments showing successful sim-to-real transfer of the learned student policy.
> - Revised writing for clarity.
>
> Please let us know if there are any additional concerns with our paper.

---

### Decision · Program_Chairs · 2021-09-13

**Decision:**

Accept (Poster)

**Comment:**

This paper proposes to combine model-free learning and model-based control to enable a quadruped robot to jump over gaps using visual inputs. All reviewers agree that the paper is tackling an important and challenging control problem that could have potentially large impact. The combination of reinforcement learning and MPC is neat and could inspire follow-up works. The revision has significantly improved the quality of this paper. Original concerns about the clarity of writing, lack of baselines, and missing on-robot experiments using the student policies are sufficiently addressed in the revision. New experiments on the robots are impressive.

The PMTG baseline for gap-crossing in simulation looks a bit weird though: many times the robot steps into the gap but seems to still get the contact force to keep it balanced (e.g. 00:06s in Trotting: Relaxed TG, rear left leg; also in the first plot Trajectory Plots). This does not change any conclusion of the paper because the PMTG baseline would perform even worse. But explaining/fixing this artifact will further improve the paper quality.